# Impact of Playground Shade Structures on Ultraviolet Radiation Exposure and Physical Activity among Children at a Childcare Facility

**DOI:** 10.3390/ijerph20136306

**Published:** 2023-07-06

**Authors:** Breann Corcoran, Parveen Bhatti, Cheryl E Peters, Fabio Feldman, Maryam Darvishian

**Affiliations:** 1Prevention, Screening & Hereditary Cancer Program, BC Cancer, Vancouver, BC V5Z 1G1, Canada; bspecht@bccancer.bc.ca (B.C.); cheryl.peters1@bccdc.ca (C.E.P.); fabio.feldman@bccancer.bc.ca (F.F.); 2School of Population and Public Health, University of British Columbia, Vancouver, BC V6T 1Z3, Canada; pbhatti@bccrc.ca; 3Cancer Control Research, BC Cancer Research Institute, Vancouver, BC V5Z 1L3, Canada; 4BC Centre for Disease Control, Vancouver, BC V5Z 4R4, Canada

**Keywords:** children, skin cancer, sun protection, shade, ultraviolet radiation, physical activity

## Abstract

Skin cancer is the most common cancer in Canada, and rates continue to rise. While sunscreen and protective clothing remain critical strategies to reduce skin cancer risk, shade is generally the most effective way to control exposure. There remains a lack of data, particularly in British Columbia (BC), demonstrating the extent to which shade availability reduces ultraviolet radiation (UVR) exposure in a real-world setting and the potential impacts of shade provision on physical activity. We evaluated the impact of shade structures on UVR exposure and physical activity at a Vancouver-area childcare center with an outdoor play area with limited existing shade. 22 children, aged 3–5 years, participated in the study. Three removable shade sails were installed in the outdoor play space, and UVR and physical activity measurements were collected during the spring, summer, and fall months. Ultraviolet B (UVB) radiation data was measured using UVB dosimeters, and physical activity data was measured using accelerometers. Data were collected during each season over a total of four days—two days with shade sails installed and two days with shade sails removed. Overall, with shade installation, UVR exposures and physical activity levels among children were reduced by 50% and 20%, respectively. This study supports the use of shade sails to significantly reduce UVR exposures among preschool-aged children in BC; however, the potential for decreased physical activity from shade sails should be further explored in future research.

## 1. Introduction

Exposure to ultraviolet radiation (UVR) from the sun, particularly the shorter wavelength ultraviolet B (UVB) radiation, can damage the skin, cause burns, and trigger the development of skin cancer [1]. Children are particularly sensitive to the harmful effects of UVR, and evidence suggests that early childhood exposures to UVR are primarily responsible for the occurrence of melanoma, the deadliest form of skin cancer, later in life [2]. Given that melanoma rates in British Columbia (BC) are projected to increase by 57% by 2030 [3], improved strategies to reduce UVR exposures among children are needed.

While messaging to caretakers about the importance of sunscreen and sun protective clothing for children remains critical, there are additional means for UVR protection that are likely more effective. In particular, the provision of shade in outdoor play spaces is an intervention that could result in significant reductions in UVR exposure among children [4]. For instance, in a US study by Vanos et al., the presence of artificial shade significantly reduced UVB_280–315nm_ exposures by 91% among children aged 9–13 years [5]. Similarly, a 2019 study among 48 Early Childcare Education Centers and 678 children from Western Australia between the ages of 3–5 years found that natural shade, specifically center vegetation, was significantly associated with decreased UV radiation exposure [6]. Although shade design as a strategy for skin cancer prevention has been reported in the above-mentioned studies, its impact has not been assessed in the context of weather and UV radiation patterns in British Columbia (BC).

The potential impacts of shade in outdoor spaces on physical activity are also important considerations. For example, shade may enhance thermal comfort and promote increased physical activity levels during the summer, but during the spring and fall months, when UVR levels are still of concern, shade structures may lead to decreased levels of physical activity due to reduced ambient light levels and temperatures in the play space. The current literature about the association between shade and physical activity level is mixed. While some studies have found that children engage in less physical activity under shade [6,7], others have found a positive association between shade and moderate-to-vigorous physical activity (MVPA) [8]. We sought to test the impacts of shade structures on UVR exposures and physical activity levels in a BC setting among children aged 3–5 years.

## 2. Materials and Methods

### 2.1. Study Site

In partnership with the City of Vancouver, the Collective Tennyson Childcare Centre (CTCC) was selected as the study site. The CTCC is a private childcare center that offers four programs: Infant Childcare, Toddler Childcare, 3–5-year-old Childcare, and Out-of-School Care. The study took place in the 3–5-year-old program’s outdoor play area, which was selected due to its south-west orientation and high levels of sun exposure. The CTCC playground is located in an urban residential, mid- to high-income neighborhood on the 2nd floor. Although situated in an urban environment, there are no buildings in its vicinity that provide shade over the playground. The full-time 3–5 program operates 5 days a week from 7:30 a.m. to 5:30 p.m. with a group size of 25 children.

### 2.2. Shade Sails

Three shade sails were installed by Shade Sails Canada, which has been designing and installing residential and commercial shade structures across Canada since 1999. The project team worked in collaboration with City staff, childcare center operators, and health licensing to determine the ideal locations for the shade sails, which were placed near the climbing structure, where children spent the majority of their outdoor time, as well as over a seating area (Figure 1). Shade sails are based on the concept of spanning tensioned fabric between perimeter posts. The posts are permanent, while the tensioned UVR stabilized fabric, which is weather-resistant and environmentally safe, can be removed and stored (e.g., during the winter months). With hooks already in place along the outdoor perimeter fence at CTCC, the shade sails did not require posts to be installed. This lowered project costs, and the total cost for all three shade sails was approximately $8000. During each of the spring, summer, and fall seasons, children at the daycare center were fitted with UVR and physical activity monitors for four days with similar weather forecasts: two days during which the shade structures were installed and another two days during which the structures were removed.

### 2.3. Study Participants

Parental consent was required to enroll children in the study. We sought parental consent at two different times throughout the course of the study—once in May 2021 before the study began and then again in September 2021 to recruit incoming children and families to the program. On each study day, participating children were fitted with lightweight UV and physical activity monitors during a scheduled mid-day playtime session. When possible, the same children were monitored during each of the four measurement days per season. Parents and staff were instructed and reminded not to alter any sun protection behaviors in response to the presence of the shade structure (e.g., application of sunscreen to their children, outfitting their children in sun protective clothing, etc.). The remaining children were given “mock” devices to wear so that they did not feel excluded from activities.

### 2.4. UVR and Physical Activity Measurements

The primary outcome measures for this study were sun exposure and physical activity. To measure sun exposure, we collected measurements of Ultraviolet B, which has a shorter wavelength and is associated with skin burning. To measure physical activity, we measured minutes of moderate to vigorous intensity of physical activity.

UVB measurements were collected using a NIWA UVB dosimeter [9] worn on the wrist of each child for a 1.5–2 h period. The electronically logged data were used to calculate equivalents of standard erythemal dose (SED), where 1 SED = 100 Joules per square meter of skin (J/m^2^) [10]. The SED is independent of skin type, such that a dose of SED may cause erythema (i.e., reddening) in some people but not all. Data on skin type and ethnic background was not collected in this study.

Physical activity was measured using a lightweight accelerometer (ActiGraph GT3X+) that was worn on the right hip. The logged vertical movement data were used to calculate minutes of moderate to vigorous intensities of physical activity (MVPA), which were defined as a minimum of 500 counts of vertical movement per minute [11]. These counts correspond to ≥3 Metabolic Equivalents (METs), which is the ratio of a person’s working metabolic rate compared to their resting metabolic rate [11].

### 2.5. Statistical Analysis

Separate generalized linear regression models (SAS Version 9.4) were used to examine the impact of shade structure installation on SED and MVPA. Both SED and MVPA data were ln-transformed to ensure normality of the data distributions, and the models were adjusted for season (spring, summer, fall), sex, age (continuous), and duration of the play period (continuous). The models also accounted for repeated observations within each season and among children. Point estimates and 95% confidence intervals (CI) for associations with SED and MVPA were exponentiated, thereby indicating the ratio of SED and MVPA when comparing the presence of shade to the absence of shade. Associations were deemed statistically significant if *p*-values were ≤0.05.

## 3. Results

Twenty-two children aged 3–5 years participated in the study, nine females and thirteen males. Children spent on average between 75 and 120 min outdoors during their morning play time period between 10 am and noon. Table 1 lists the dates, weather forecasts, maximum temperatures, maximum UV index (UVI), number of children measured, and duration of play periods for each of the study measurement days. The spring data collection period was unseasonably warm, with temperatures reaching up to 25 °C compared to average highs of 20 °C (Government of Canada) [12]. The maximum UVI reached 6.1 during the spring data collection period. By contrast, the summer data collection period saw temperatures up to 22 °C with a maximum UVI of 4.9.

Overall, the mean SED was lower when shade sails were installed (0.2) as compared to when the sails were absent (0.5) (Table 1). This effect was consistent across each of the seasons. The highest SED levels were observed during the spring as compared to the summer, which is likely attributable to higher UV indices for the spring versus summer measurement days (Table 1). While the overall mean minutes of MVPA were lower when shade sails were installed, this was not consistent across the seasons; mean minutes of MVPA were higher with shade installation in the summer but lower with shade installation in the spring and fall (Table 2).

In the regression analyses, after adjusting for season, sex, age, and playtime duration, the SED with installation of the shade sails was 0.5 (95% CI: 0.4, 0.7, *p* < 0.0001) of the SED when shade sails were not installed (Table 3). This indicates a statistically significant 50% reduction in harmful UVR exposure due to the installation of the shade sails. The MVPA with installation of the shade sails was 0.8 (95% CI: 0.6, 0.9, *p* = 0.007) of the MVPA when shade sails were not installed. This indicates a statistically significant 20% reduction in moderate to vigorous levels of physical activity due to the installation of the shade sails (on average, equivalent to a 10-min reduction in MVPA). As a sensitivity analysis, the MVPA model was rerun after removing a potentially outlying MVPA measurement (2.0 min). The results of this analysis were nearly identical.

## 4. Discussion

The study demonstrated that the installation of shade sails significantly reduced harmful UVR exposures among young children. Without the shade sails, SEDs of up to 2.5 were observed. With the installation of the shade sails, the maximum SED observed was 0.8. There is no well-defined recommended maximum daily dose of SED for children since any amount of exposure to SED could potentially be linked to skin cancer. These results add to an existing base of literature showing the benefit of shade sails in reducing UV exposure among children [5,13,14]. For example, a 2017 study conducted in Texas found that shade significantly reduced personal UV erythemal exposures by 55% [5]. Many of these studies have largely taken place in Australia, New Zealand, and the United States (areas with much higher ambient UV radiation), so our study offers important local context for BC’s climate. BC experienced some of the hottest weather on record in 2021, so it is increasingly important for us to consider the impacts of climate change and its effects on our children and communities [15]. Recent literature has shown that an increase of 2.5 °C in the annual average temperature is associated with an increase in the number of expected melanoma incidences by 21.1% in a given region [16].

Unlike previous studies, which have demonstrated either no impact of shade on physical activity levels or a positive impact of shade on physical activity levels [4,6,17], our study found that the installation of shade sails reduced MVPA. Specifically, the shade sails were associated with lower levels of physical activity during the warmest and coolest seasons; temperatures were not assessed at the study site but were taken from local weather forecasts. By comparison, Boldemann et al. measured physical activity among 4–6-year-old preschool students using pedometers (step count/minute) and found that preschool environments with natural shade features triggered physical activity [4]. Similarly, a study involving nearly 500 Denver residents showed a positive association between the presence of shade and trees and reported levels of moderate-to-vigorous physical activity; these results, however, were not specific to preschool-age children. Another study among preschoolers in Western Australia aged 3–5 years did not find any significant association between shade and physical activity but did find a positive association between shade and increased time spent outdoors [6]. The exact reason for our discrepant findings is unknown, though, it may be, for example, attributable to differences in playtime programming across data collection days. The provision of shade can promote time spent outdoors for children attending childcare centers [6]. It may be that children engage in more physical activity since they are outdoors longer, allowing for greater time to accumulate minutes of physical activity, as a systematic review on the topic suggests [18], but this warrants further investigation. The positioning and location of the shade sails at park and playground locations should also be further examined. In many cases, children are found to be playing in direct sunlight, while parents, educators, and caregivers rest in shaded areas [7]. A German study analyzed the percentage of shaded areas compared with the total playground area and the main play area for children in 144 playgrounds and found that, on average, 41% of the total playground area was shaded, while only 22% of the main play area for children was in the shade [7]. A focused design of children’s recreational spaces to utilize shade is essential to reducing UVR exposures during play and extending safe outdoor stays.

Future research on the effectiveness of natural and/or portable shade designs for UV protection is needed. While shade sails can be highly effective for sun protection, the cost to manufacture and install the shade sails may be prohibitive for some childcare centers. Lower-cost alternatives to shade sails are therefore needed to add to the inventory of potential shade interventions for centers to consider. Additionally, natural shade can have the dual benefit of combating the urban heat island effect and should be prioritized when possible. Another potential challenge with the use of shade sails is the ongoing maintenance and care required. For example, if left outdoors year-round, the shade sails will weather more quickly and have a shorter lifespan, but taking them down and reinstalling them often requires maintenance staff. Perspectives from Early Childcare Educators, center administrators, and building maintenance staff are needed to gather information on the logistical barriers and facilitators to the application of shade sails in childcare centers. Shade sails are growing in popularity but are still not widely used in British Columbia, so more research is needed to understand how they perform in BC’s climate and some of the unique logistical considerations. The strengths of the study included capturing quantitative UVR and physical activity data from individual children over multiple seasons. Unfortunately, we were unable to outfit the same children with measurement devices consistently throughout the study. This was due to turnover during the fall data collection period as children transitioned to kindergarten. Additionally, there was variability in the number of children that attended over the summer months due to summer vacation. This contributed to the variability in play behavior among study participants, which may have affected results. While we did our best to select measurement days with similar weather forecasts, there was some variability in weather between days within a season, depending on whether the shade sails were installed or not. Data collection took place in the morning, but since the site was south-west exposed, afternoon data collection may have better reflected peak sun exposure and UV levels that children and staff experienced on site.

Another important factor to note is the potential drawback that the UVB dosimeters were worn on the wrists of children. This is one of the most mobile parts of the body, thus subject to variability. Additionally, the device may have at times been covered by jackets or long-sleeve clothing worn by the children, and measurements of UVB exposure may therefore be an underestimate. Finally, the wrist is not representative of face exposure because the face is vertically oriented toward the sun.

As the current study was a pilot, the sample size was not large, and the study was limited to one childcare facility in Vancouver. Hence, we captured measurements of 7–8 children at a time and used repeated observations over a period of four days in each of the three seasons (i.e., 12 days and a total of 89 individual observations). To increase the study’s generalizability, future research should include more childcare centers and a broader age range. This pilot study, however, provides the required evidence for future research and offers strong quantitative results to support the use of shade sails for skin cancer prevention. More research is needed to understand how shade affects playtime duration, thermal comfort, and associated physical activity among children playing outdoors. UV levels are not related to temperature, so outdoor thermal comfort is important to consider when measuring physical activity. Future studies should focus on measuring ambient temperatures and understanding how children perceive their thermal comfort during outdoor play.

## 5. Conclusions

Study results showed a significant impact of shade sails on UVR, supporting the use of these structures to reduce harmful exposures. Temporary shade structures or removable shade sails offer flexibility for use during high temperatures and UV levels and removal with lower UV levels and temperatures to allow for solar access and increased temperatures during cooler periods. Natural shade via deciduous plants and trees may also be a good solution, with a canopy and plants blooming during peak spring/summer months and offering UV protection. Results also indicated that shade sails may negatively impact physical activity levels, though the exact reason for this could not be discerned. Future studies should include measurement of ambient temperatures at the study site and observation of play behaviors to identify strategies that can mitigate the negative impacts of shade on physical activity.

## Figures and Tables

**Figure 1 ijerph-20-06306-f001:**
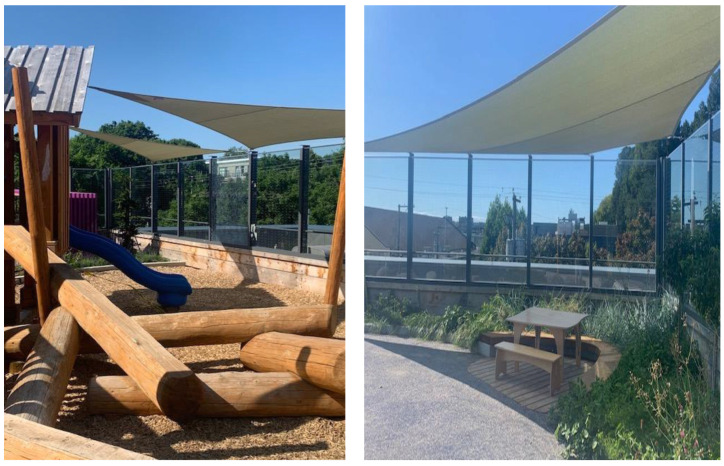
Photos of the three shade sails that were installed at CTCC.

**Table 1 ijerph-20-06306-t001:** Study dates, weather forecasts, maximum temperatures, maximum UV index (UVI), number of children measured, and duration of play periods for each of the study measurement days.

Season	Date	Weather Forecast	Max Temp (°C)	Max UVI	Shade	Children Measured	Playtime Duration (min)
**Spring**	02-June	Sun with clouds	25	6.7	No	8	90
**Spring**	04-June	Partly sunny	18	3.5	No	8	120
**Spring**	17-June	Partly sunny	18	5.9	Yes	8	105
**Spring**	24-June	Partly sunny	22	5.1	Yes	8	90
**Summer**	25-August	Passing clouds	22	1.7	Yes	7	105
**Summer**	13-September	Partly sunny	17	2.7	Yes	7	90
**Summer**	15-September	Partly sunny	16	4.9	No	8	75
**Summer**	16-September	Passing clouds	15	4.6	No	7	75
**Fall**	04-October	Broken clouds	14	3.3	No	7	90
**Fall**	07-October	Partly sunny	11	3.4	No	7	75
**Fall**	29-October	Passing clouds	10	2.6	Yes	7	90
**Fall**	01-November	Passing clouds	8	3.4	Yes	7	90

**Table 2 ijerph-20-06306-t002:** Summary statistics for standard erythema dose (SED) and minutes of moderate to vigorous physical activity (MVPA), shade versus no shade, overall and by season.

	Standard Erythema Dose				Minutes of Moderate to Vigorous Physical Activity			
	Shade		No Shade		Shade		No Shade	
**Season**	Mean (SD) *	Min/Max	Mean (SD) *	Min/Max	Mean (SD) *	Min/Max	Mean (SD) *	Min/Max
**Spring**	0.4 (0.2)	0.1/0.8	1.0 (0.6)	0.2/2.5	48.1 (15.3)	22.0/69.0	57.1 (14.1)	23.0/75.0
**Summer**	0.1 (0.1)	0.0/0.4	0.2 (0.1)	0.1/0.5	47.9 (15.7)	21.0/72.0	40.5 (8.8)	26.0/52.0
**Fall**	0.1 (0.1)	0.02/0.3	0.3 (0.4)	0.1/1.5	35.5 (13.1)	2.0/55.0	52.1 (12.4)	25.0/72.0
**Overall**	0.2 (0.2)	0.0/0.8	0.5 (0.6)	0.1/2.5	44.0 (15.6)	2.0/72.0	50.0 (13.7)	23.0/75.0

* SD—Standard Deviation.

**Table 3 ijerph-20-06306-t003:** Overall effects of shade installation on standard erythema dose (SED) and minutes of moderate-to-vigorous physical activity (MVPA).

Shade Yes (Ref No)	Estimate (95% CI) *	*p*-Value
Standard erythema dose	0.5 (0.4, 0.7)	<0.0001
Minutes of moderate to vigorous physical activity	0.8 (0.6, 0.9)	0.007

* Adjusted for season, sex, age, and playtime duration.

## Data Availability

The data presented in this study are available on request from the corresponding author. The data are not publicly available due to ethical restrictions.

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
