# Peer review of "Impact of Playground Shade Structures on Ultraviolet Radiation Exposure and Physical Activity among Children at a Childcare Facility"

_ijerph, 2023, doi:10.3390/ijerph20136306_

Round 1

Reviewer 1 Report

I enjoyed reading this paper.  

Some suggestions.

In the abstract  there should be some reference to the risk of tradeoff, eg. better sun protection at the risk of less physical activity.  This may be a "one off" and as the authors note, they cannot explain it..

In intro, they note that while this shade studies have been done elsewhere they have not been done in BC. (this is mentioned in the discussion but putting a sentence about this in the intro would be helpful as well)

The study is done at a daycare center -- who attends? is it public or private?

Do the authors know how much do this shade sails cost?  In the discussion might discuss how available these are and whether they could be adapted to many types of parks/playgrounds. 

For participants -- the authors . On each study day, participating children were fitted with lightweight UV and physical activity monitors during a scheduled mid-day playtime session -- 

How many children attend the daycare center and of those how many participated.

2.4 clearly state what the relevant  outcome measures for the study, eg. sun exposure and physical activity

I got a little lost in how the study was set up and I wonder if the authors might add a flow chart to make it clearer -- 

Were there enough children in the study to see if there were any differences between boys and girls for the outcome measures? 

Is there any information about the children's ethnicity.

Other limitations is that this is a small sample.  Also I agree about the concern about generalizability -- any counterpoint about why it is still useful

I think the lack of physical activity is a concern as the authors note and I understand that they cannot explain it -- however do they have suggestions if the effect is real about how to minimize it.

They mention in the discussion the flexibility of putting these sails up -- how long to they take to put up?  How widely used are they?  So maybe expand a couple of sentences about this in the discussion.

Any suggestions for follow up research?

Author Response

Reviewer 1:

Comments and Suggestions for Authors

I enjoyed reading this paper.  

Some suggestions.

  1. In the abstract, there should be some reference to the risk of trade-off, e.g. better sun protection at the risk of less physical activity.  This may be a "one off" and as the authors note, they cannot explain it.

Response:

We now included statement in abstract about the potential for decreased physical activity as trade off.

  1. In intro, they note that while this shade studies have been done elsewhere they have not been done in BC. (this is mentioned in the discussion but putting a sentence about this in the intro would be helpful as well).

Response:

We now included a sentence to explain that this research has not been conducted in BC.

  1. The study is done at a daycare center -- who attends? is it public or private?

Response:

Indeed the study is conducted at a childcare centre in Vancouver; we added more detail in the study setting section of the article.

  1. Do the authors know how much do this shade sails cost?  In the discussion might discuss how available these are and whether they could be adapted to many types of parks/playgrounds.

Response: 

Approximate cost is now included. 

  1. For participants -- the authors. How many children attend the daycare center and of those how many participated.

Response:

25 children attend the daycare setting – we now included the site description in the paper.

  1. clearly state what the relevant outcome measures for the study, eg. sun exposure and physical activity.

Response:

Detailed are now included in section 2.4 of the manuscript.

  1. I got a little lost in how the study was set up and I wonder if the authors might add a flow chart to make it clearer -- 

Response:

The following flow chart is now included:

  1. Were there enough children in the study to see if there were any differences between boys and girls for the outcome measures? 

Response:

In this pilot study in total 9 girls and 13 boys were included. The generalized linear regression models accounted for repeated observations within each season and within children. Considering the study design and the repeated measures, we could adjust the models for the season, sex, age, and playtime duration.

  1. Is there any information about the children's ethnicity?

Response:

This information was not collect. We now included a sentence about this in section 2.4.

  1. Other limitations is that this is a small sample.  Also I agree about the concern about generalizability -- any counterpoint about why it is still useful

Response:

More detailed on study limitations is now included in the discussion section.

  1. I think the lack of physical activity is a concern as the authors note and I understand that they cannot explain it -- however do they have suggestions if the effect is real about how to minimize it.

Response:

            Our study was a pilot and in the discussion section we provided a recommendation for the necessity of conducting similar studies in the furure including more childcare facilities and children with broader age range to further investigare the effect of shade on level of physical activity.

  1. They mention in the discussion the flexibility of putting these sails up -- how long to they take to put up?  How widely used are they?  So maybe expand a couple of sentences about this in the discussion.

Response:

We now added a paragraph in the discussion on this issue.

  1. Any suggestions for follow up research?

Response:

We now included some suggestions for future research in discussion.

Reviewer 2 Report

General comment

This paper address the issue of shade in outdoor play spaces, which is an important factor in skin cancer prevention among children. The article is well written and easy to read. The main limitation is the fact that the number of participants-day is small, which limits the dataset to a reduced number of meteorological (cloudiness, temperature) and situational (shade or no shade) conditions. That said, the design is well thought and the study brings some additional evidence of the effectiveness of shading on UVR protection in playgrounds. The observed increase on physical activity is difficult to interpret in the context of this study though.

Specific comments

Abstract. Provide a short explanation about the data collection method (measurements…)

§2. Material and methods. Wrist UVR measurement is fairly common and often easier to implement in than other locations. The drawback is that it is also one of the most mobile body part (and thus a large variability) and it is not very representative of face exposure (which is one of the main issue), because the face is vertically oriented. There is nothing wrong with this choice, but the author should perhaps emphasize these limits in the discussion. 

§3. results Section: this section refers to 22 participants, while there is only 21 reported in the Abstract

§ Discussion. About the discrepant finding reported for physical activity. I am not really surprised here. In my understanding, physical activity is reduced due to thermal stress (or at least thermal discomfort). Unlike UVR, which remains an issue for a wide range of weather conditions, thermal discomfort is an issue during hot days. Since the temperatures appear to be very mild in Canada, I don’t expect this effect to be significant. Maybe the author should put their results in perspective with the observed thermal conditions in the studies showing a decrease in physical activity.

Author Response

Reviewer 2:

  1. General comment

This paper address the issue of shade in outdoor play spaces, which is an important factor in skin cancer prevention among children. The article is well written and easy to read. The main limitation is the fact that the number of participants-day is small, which limits the dataset to a reduced number of meteorological (cloudiness, temperature) and situational (shade or no shade) conditions. That said, the design is well thought and the study brings some additional evidence of the effectiveness of shading on UVR protection in playgrounds. The observed increase on physical activity is difficult to interpret in the context of this study though.

 Response:

Specific comments

  1. Provide a short explanation about the data collection method (measurements…)

Response:

 We now added a sentence in the abstract about the data collection method.

  1. Material and methods. Wrist UVR measurement is fairly common and often easier to implement in than other locations. The drawback is that it is also one of the most mobile body part (and thus a large variability) and it is not very representative of face exposure (which is one of the main issue), because the face is vertically oriented. There is nothing wrong with this choice, but the author should perhaps emphasize these limits in the discussion. 

 Response:

We now added this limitation in the discussion.

  1. Results Section: this section refers to 22 participants, while there is only 21 reported in the Abstract.

Response:

We adjusted the abstract accordingly.

  1. About the discrepant finding reported for physical activity. I am not really surprised here. In my understanding, physical activity is reduced due to thermal stress (or at least thermal discomfort). Unlike UVR, which remains an issue for a wide range of weather conditions, thermal discomfort is an issue during hot days. Since the temperatures appear to be very mild in Canada, I don’t expect this effect to be significant. Maybe the author should put their results in perspective with the observed thermal conditions in the studies showing a decrease in physical activity.

Response:

We now includeda a sentence in the discussion to acknowledge that thermal stress is different than UV radiation and that thermal comfort should also be considered in the context of physical activity measures.
